# Effects of Chemotherapy and Radiotherapy on Spermatogenesis: The Role of Testicular Immunology

**DOI:** 10.3390/ijms20040957

**Published:** 2019-02-22

**Authors:** Ning Qu, Masahiro Itoh, Kou Sakabe

**Affiliations:** 1Department of Anatomy, Division of Basic Medical Science, Tokai University School of Medicine, Kanagawa 259-1193, Japan; sakabek@tokai-u.jp; 2Department of Anatomy, Tokyo Medical University, Tokyo 160-8402, Japan; itomasa@tokyo-med.ac.jp

**Keywords:** male infertility, cancer, testicular immunology, azoospermia

## Abstract

Substantial improvements in cancer treatment have resulted in longer survival and increased quality of life in cancer survivors with minimized long-term toxicity. However, infertility and gonadal dysfunction continue to be recognized as adverse effects of anticancer therapy. In particular, alkylating agents and irradiation induce testicular damage that results in prolonged azoospermia. Although damage to and recovery of spermatogenesis after cancer treatment have been extensively studied, there is little information regarding the role of differences in testicular immunology in cancer treatment-induced male infertility. In this review, we briefly summarize available rodent and human data on immunological differences in chemotherapy or radiotherapy.

## 1. Introduction and Immunological Tolerance of the Testis

The testis is an immune-privileged site at which immunogenic germ cells are protected from the detrimental effects of immune responses. The most commonly recognized mechanism for the immunological privilege is the blood–testis barrier (BTB), physically formed by the borders of adjacent Sertoli cells, limiting the access of germ cell antigens to interstitial immune cells and the passage of antibodies from the interstitium to the tubular lumen. Immunological privilege outside the BTB involves secretion of immunosuppressive factors mainly by macrophages, Sertoli cells, peritubular cells, and Leydig cells [1,2,3,4,5] (Table 1). 

The testicular capacity to inhibit local immune responses has been confirmed in many studies; however, the mechanisms through which immune-privilege contributes to protection of spermatogenesis, a highly specialized process, have not been clearly defined. Elevated levels of intratesticular testosterone and/or progesterone may cause inhibition of local immune responses [6]. However, the testes contain high levels of steroidal molecules that are immunosuppressive by nature [7]. The expression of functional FasL by Sertoli cells [8] and/or by germ cells [9] as an active mechanism induces cell death via apoptosis, such as activated T cells in inflammation [10,11]. Furthermore, various anti-inflammatory cytokines, such as interleukin-10, are abundantly produced by testicular cells, particularly testicular macrophages [12], and members of the transforming growth factor-β family are highly expressed by Leydig and Sertoli cells [13]. The immune environment in the testes must be tightly controlled to maintain immune homeostasis for normal spermatogenesis. Disruption of immune homeostasis may result in autoimmune or infectious aspermatogenesis, thereby impairing testicular function. The mechanisms underlying the autoimmune inflammatory response and the mechanisms of orchitis have been broadly investigated and comprehensively reviewed in rats and mice [14,15,16]. 

The growing population of young cancer survivors and the trend toward postponing pregnancy until later in life have shifted researchers’ focus toward understanding treatment-induced sequelae, particularly the effects of cancer and/or treatment on fertility [17]. Although prepubertal testes do not undergo spermatogenesis and do not produce mature spermatozoa, the testes are sensitive to cytotoxic drugs and irradiation at this age. Testicular damage is drug specific and dose related, and the recovery of spermatogenesis varies following cytotoxic insults; additionally, the extent and speed of recovery are related to the agent used and the dose received [18,19,20,21]. Radiotherapy-induced testicular damage is similarly dose dependent, with speed of onset, chance of reversal, and time to recovery of spermatogenesis all related to the testicular dose of irradiation [17,22]. 

In this review, we aimed to outline immunological differences in cancer treatment-induced male infertility. Knowledge of this immunopathologic microenvironment will be useful to understand infertility and gonadal dysfunction as adverse effects of anticancer therapy.

## 2. Chemotherapy

Many drugs, particularly alkylating agents, have been shown to be gonadotoxic. Indeed, some chemotherapies used in the treatment of lymphoma or in preparation for bone transplantation have been shown to cause azoospermia, with feedback-raised follicle-stimulating hormone (FSH) levels in over 90% of men following cyclical chemotherapy [22,23].

Busulfan is a chemotherapeutic agent that is used to treat various malignant diseases, such as chronic myeloid leukemia and polycythemia vera [24,25]. Additionally, busulfan is also commonly used prior to hematopoietic stem cell transplantation [24]. Unlike other chemicals that destroy differentiated spermatogonia, busulfan is a potent agent that preferentially kills spermatogonial stem cells [26] and does not have any effect on DNA synthesis. Consequently, busulfan treatment is the most common method used to prepare recipients of spermatogonial stem cells transplantation [27,28] and to study spermatogonial stem cells kinetics and fertility recovery [29,30,31]. Busulfan can eliminate almost all endogenous germ cells in the recipient, creating an empty space in the spermatogonial stem cells niche; therefore, this drug has been used successfully to prepare recipients in mice [32,33], rats [34,35], pigs [36,37], dogs [38], hamsters [39], monkeys [40], and roosters [41]. 

Busulfan treatment has been shown to disrupt spermatogenesis by damaging germ cells and Sertoli cells [42,43]. Although busulfan has been extensively studied regarding to induction of testicular damage through various cellular and molecular mechanisms, the effects of treatment on testicular immunology have not been thoroughly evaluated (Table 1). Choi et al. showed that busulfan may induce germ cell apoptosis through loss of c-kit signaling in a Fas/FasL- and p53-independent manner in 8–12-week-old mice [44]. Moreover, using the same adult mice, Li et al. demonstrated that p53 is a key protein with roles in busulfan-induced apoptosis through reactive oxygen species (ROS)-dependent activation of the extracellular signal-regulated kinase/p38 pathway, and decreased concentrations of deacetylated p53 result in spermatogonial cell resistance to apoptosis [45]. Xian et al. showed that spermidine/spermine N1-acetyltransferase 2 (Sat2) is present in adult mice testicular Sertoli cells and that its expression is significantly increased by busulfan treatment. Furthermore, upregulation of Sat2 by busulfan alters the growth and function of Sertoli cells and causes male infertility [46]. Zhang et al. demonstrated that busulfan-induced spermatogenic cell damage upregulates tumor necrosis factor (TNF) α and macrophage chemotactic protein (MCP) 1 expression in Sertoli cells and facilitates macrophage infiltration into the testes of 8–10 weeks age mice [47]. Additionally, damaged germ cells in busulfan-treated mice release endogenous Toll-like receptor (TLR) ligands to activate TLR2 and TLR4 in Sertoli cells, thus initiating endogenous inflammation in the testes [47,48]. Activation of TLRs induces inflammatory gene expression, which may facilitate injury repair and lead to further pathological conditions, such as autoimmune diseases [49,50,51]. 

Taken together, these studies suggested that the key factors mediating testicular immunology in busulfan-induced aspermatogenesis are Sertoli cells and macrophages, being similar with that in autoimmune orchitis excluding inflammatory reactions (Table 1, Figure 1).

Sinisi et al. investigated the occurrence of antisperm antibody (ASA) in 264 prepubertal male boys (ages 1.2–13 years) treated with chemotherapy and confirmed that 26 of these patients were ASA-positive [52]. Moreover, of the 26 ASA-positive boys, 24 had genital tract abnormalities, such as cryptorchidism, testicular torsion, and hypospadias, and two had leukemia with testicular infiltration [52]. Therefore, these findings suggested that chemotherapy did not modify antibody positivity. Only one experiment showed that remaining spermatogonia after 40 mg busulfan-treatment reacted strongly to IgG antibodies and that serum IgG levels increased in a manner corresponding with the increase in testicular IgG levels in adult (8–12-week-old) ICR mice [53]. Researchers demonstrated that serum IgG increased from 4 weeks after busulfan treatment, peaked at 7 weeks, and dropped rapidly to control concentrations after 8 weeks. In contrast, the testicular levels of IgG showed a gradual increase that accelerated after 3 weeks and peaked at 6 weeks, before dropping to control serum levels at 8 weeks. In our previous busulfan study in 4-week-old C57BL/6 mice, we examined serum anti-germ cell antibodies in 40 mg busulfan-treated mice and demonstrated that no anti-germ cell antibody production could be detected at 60 days after busulfan treatment (according to immunohistochemistry using serum samples and anti-mouse IgG) [48]. Moreover, we demonstrated that busulfan treatment progressively decreased the weight of the testes and the epididymal sperm count from day 60 to 120 and that 40 mg busulfan-induced aspermatogenesis was irreversible for at least 360 days [48,54,55]. We surmised that the primary cause of this aspermatogenesis involved factors other than ASA. We also showed that normalized macrophage migration and reduced expression of TLR2 and TLR4 after busulfan treatment could completely rescue the injured seminiferous epithelium and alleviate aspermatogenesis [48]. These differences may be related to the different experimental periods because it is well known that the infertility after busulfan is in a time- and dose-dependent manner, and is also possibly related with different species or ages of mice.

## 3. Radiotherapy

Irradiation has been shown to decrease spermatogenesis, alter the production of various hormones, and induce infertility. In both rodents and humans, the extent of testicular injury is directly related to the dose of irradiation delivered [56,57], and the germinal epithelium is very sensitive to radiation-induced damage [58], with changes in spermatogonia following doses as low as 0.1 Gy and permanent infertility after fractionated doses of 2 Gy and above [59].

Damage may be caused during direct irradiation of the testes or from scattered radiation during treatment directed at adjacent tissues. No recovery of spermatognesis was observed in 10 patients, mean age 32.9 years (range 24–40), receiving doses of 1.4–2.6 Gy after 17–43 months follow-up; however, recovery of fertility was observed in two patients with testicular radiation doses of 1.2 Gy and recovery of spermatogenesis was observed in all eight patients who received radiation doses of 0.28–0.9 Gy for testicular seminoma [60]. In postpubertal men, testicular doses of less than 0.2 Gy have no significant effect on FSH levels or sperm counts, whereas doses between 0.2 and 0.7 Gy cause transient dose-dependent increases in FSH and a reduction in sperm concentrations [61]. Notably, total body irradiation (TBI), used for bone marrow transplantation, is associated with appreciable gonadal toxicity, and previous studies have shown that 99.5% of men (mean age 31 years ranging from 11- to 62-years-old) who received 12.0 Gy TBI showed permanent infertility [23]. Moreover, TBI doses as low as 5–6 Gy could cause decreased spermatogenesis in the seminiferous tubules in prepubertal mice [62,63].

The deleterious effects of irradiation in biological systems are mainly mediated through the generation of ROS and cause lipid peroxidation in the cellular membrane, thereby inducing DNA damage in immature germ cells [64,65]. DNA damage caused by irradiation in premeiotic germ cells is detectable in primary spermatocytes and is still present in mature spermatozoa [64,66]. Furthermore, apoptosis of germ cells has been reported as a mechanism responsible for infertility in irradiated testes [67,68]. Some studies have indicated that irradiation-mediated oxidative stress induces apoptosis primarily in adult mouse and rat germ cells [69,70,71,72]. Additionally, irradiation-induced germ cell apoptosis depends on activation of caspase-3 in three-month-old rats [72], concomitant with increased expression of caspase-8 and decreased expression of caspase-9 in adult rats and prepubertal mice, respectively [72,73] (Table 1, Figure 1). However, irradiation-induced apoptosis does not occur in all types of testicular somatic cells. Some studies have shown that Sertoli cells and Leydig cells are resistant to irradiation-induced apoptosis [67,71,74,75,76], whereas some studies have described minimal changes in Sertoli cells [77,78].

Sertoli cells, through formation of the BTB, protect postmeiotic germ cells from exogenous toxicants introduced by testicular blood and lymph. Recently, some studies have demonstrated that Sertoli cell junctional proteins are the primary cellular targets of reproductive toxicants, such as cadmium chloride, dichlorodiphenyltrichloroethane, cisplatin, and bisphenol A [79,80,81]. In contrast, some reports have shown irradiation-induced BTB disruption with a decrease in zonula occludens-1 (ZO-1), occludin, and/or claudin-11 [82,83,84]. The integral membrane proteins claudin-11 [85,86] and occludin [87,88,89] and the adaptor protein ZO-1 [90,91], which all function in tight junctions, are critical components of the BTB. In both prepubertal and adult claudin11-knockout mice, the lumen of the seminiferous tubules are narrowed [92,93], round spermatids are the most mature germ cells, and increased germ cell apoptosis is observed [93]. In occludin-knockout mice, the seminiferous tubules are atrophied and have a Sertoli-cell-only phenotype at 40–60 weeks of age [94].

Breakdown of the BTB with its subsequent leukocytic infiltration of tubules can be found in human biopsy specimens from cases of idiopathic infertility and in models of testicular inflammation [95,96,97] (Table 1). An increase in BTB permeability is known to enhance ASA production, resulting in infertility in males [85,86,98]. Recently, we reported that single-dose TBI induced ASA that preferentially reacted with mature spermatids and spermatozoa in prepubertal mice [73]. Our results showed that 6 Gy of TBI induced a disruption of spermatogenesis with a decrease in inter-Sertoli tight junction mRNA levels and the production of ASA (Table 1). The contributions of claudin-11, occludin, and ZO-1 to BTB integrity were further determined in TBI-induced aspermatogenesis, and recovery of spermatogenesis was found to depend on the recovery of the above disorganized tight junctions. Furthermore, differences in busulfan- and irradiation-induced aspermatogenesis and the treatment duration in response to busulfan- and irradiation-induced aspermatogenesis were evaluated. Busulfan treatment in the same prepubertal mice was found to progressively decrease the weight of the testes and the epididymal sperm count from day 30 to day 120 (Figure 2); whereas an administered an oriental medicine completely rescued these effects on day 120 after busulfan treatment (testes weight: 0.100 ± 0.006 g; epididymal spermaotzoa: (21.680 ± 1.700) × 10^5^ cells) [48]. In contrast, irradiation treatment induced significant decreases in the weights of the testes and epididymal sperm cell counts on day 30; marginal recovery was observed from day 60 to day 120, but further decreases in all parameters were noted on day 150 (Figure 2). Notably, supplementation with the above oriental medicine significantly restored the epididymal spermatozoa count and fertility on day 150 but not day 120 [73]. This belated recovery of spermatogenesis in the irradiated group compared with that in the busulfan-treated group suggested that the presence of ASAs may be the other immunological cause of aspermatogenesis. Although no leukocytic infiltration was detected in the irradiated testes, the breakdown of BTB and the immune responses against the testicular autoantigen are similar to that in the orchitis testes (Table 1).

## 4. Conclusion

From the above limited reported data, the impaired reproductive functions induced by cancer treatment including chemotherapy and radiotherapy are related with the different immune-pathophysiological conditions. Especially, with the breakdown of BTB by irradiation, germ cell autoantigens inside BTB might leak out repeatedly, leading to a continuous supply of autoantigens for immune stimulation with resultant ASA production and prolongation of the testicular inflammation. It is well known that the mechanisms of immune privilege in male reproductive organs are still far from being completely understood. The mechanisms underlying the autoimmune inflammatory response and the mechanisms of orchitis have been broadly investigated and comprehensively reviewed in rodent and human. Although damage to and recovery of spermatogenesis after cancer treatment have been extensively studied, there is little information regarding the role of differences in testicular immunology in cancer treatment-induced male infertility. Presently, the information on therapy for cancer treatment-induced male infertility is also limited. Because treatment with cytotoxic chemotherapy and radiotherapy indicated the increased FSH levels and reduced sperm counts, the effects of suppression of testosterone and gonadotrophin analogs on stimulation of spermatogonial differentiation, resulting in spermatogenic progression after cancer treatment, have been well demonstrated [99,100]. However, hormone suppression has multiple side effects and recovery occurs gradually, the application of hormone suppression treatments to enhance endogenous spermatogenic recovery has so far been successful in clinical trials [99]. Furthermore, chemotherapy and radiotherapy are often used in combination associated with greater testicular dysfunction and germinal epithelial damage. To lead to an availability of therapy on male infertility after cancer treatment, elucidation of the immunological mechanisms underlying aspermatogenesis by cancer treatment could be helpful.

## Figures and Tables

**Figure 1 ijms-20-00957-f001:**
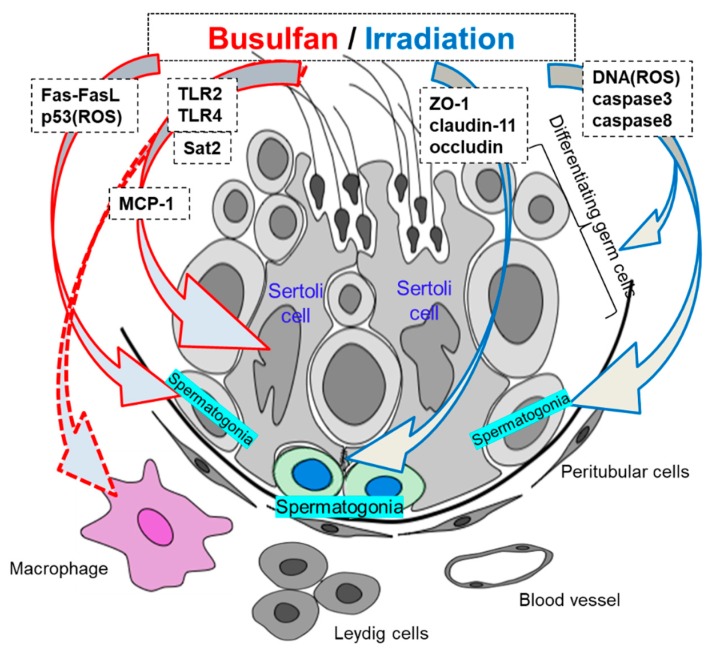
Immunological differences in the testes after cancer treatment.

**Figure 2 ijms-20-00957-f002:**
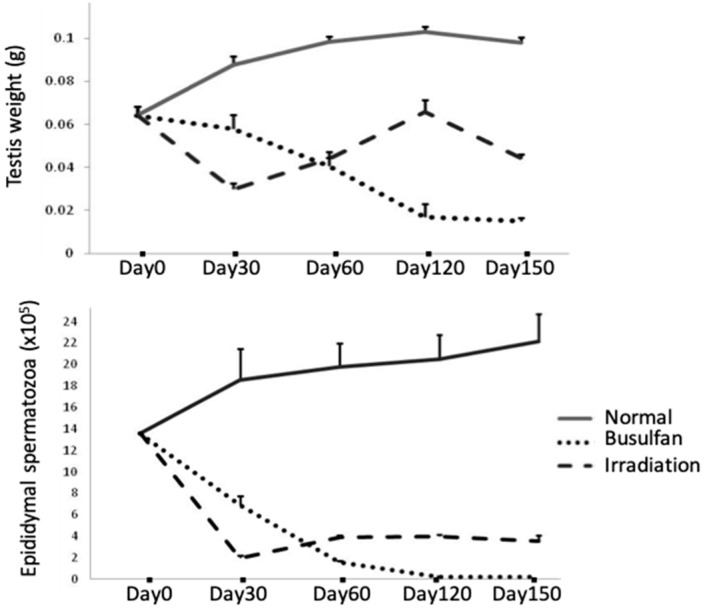
Testes weights and epididymal spermatozoa numbers in normal and busulfan-/irradiation-treated mice.

**Table 1 ijms-20-00957-t001:** Testicular immune factors in normal and cancer-treated mice.

	Immunosuppressive Factors in Normal Testis	Local Function	Testicular Immunology Consequences after Busulfan-Treatment	Testicular Immunology Consequences after Irradiation-Treatment	Testicular Immunology Consequences in Autoimmune Orchitis	
Germ cells	transforming growth factor β	Leydig cell steroidogenesis ↓				
	Fas ligand	apoptosis of Fas-bearing lymphocyte	↑ or (-)			
	interferon-γ	Leydig cell steroidogenesis ↓				
	tumor necrosis factor α	Leydig cell steroidogenesis ↓ or ↑				
				Fas ↑ caspase3-8 ↑	Fas ↑	caspase 3–8 ↑
			Fas ↑	apoptosis through oxidative stress	Bax ↑	caspase 9 ↑
			p53-ROS ↑ caspase3 ↑	DNA damage		
Sertoli cells	activin	mitogenesis of lymphocytes ↓				
	inhibin	mitogenesis of lymphocytes ↑				
	interleukin-6	meiotic DNA synthesis of germ cell		↑ ?	↑	
	Fas ligand	apoptosis of Fas-bearing lymphocyte				
	transforming growth factor β	inhibin secretion ↑				
			TNFα↑ MCP-1 ↑ TLR2,4 ↑	ZO-1, occludin, claudin-11 ↓	occludin, claudin-11 ↓	
			Sat2 ↑			
Leydig cells	testosterone		↓	↓	↓	
	protein S	Leydig cell steroidogenesis ↓				
	insulin-like growth factor-1	testosterone secretion ↑				
	Fas ligand	apoptosis of Fas-bearing germ cell				
	interleukin-10	immune privilege				
	transforming growth factor β	contractility of myoid cell				
			Leydig cell apoptosis			
Testicular	interleukin-10	inhibition of T cell-mediated immune response response				
macrophages	interferon-γ	Fas ligand expression by Sertoli cell ↑			↑	
	interleukin-6	radioprotection of germ cell by Sertoli cell		↑ ?	↑	
	tumor necrosis factor α	Fas ligand expression by Sertoli cell ↓ or ↑	↑		↑	
			macrophage infiltration (+)	macrophage infiltration (-)	macrophage infiltration (+)	
Others			ASA?	ASA (+)	T cells·B cells infiltration (+)	
					ASA (+)	

↑ indicated increase and ↓ indicated decrease; ? indicated different opinion.

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
