# Peer review of "Effects of Chemotherapy and Radiotherapy on Spermatogenesis: The Role of Testicular Immunology"

_ijms, 2019, doi:10.3390/ijms20040957_

Reviewer 1 Report

In this review, the impact of the administration of busulfan (a chemotherapeutic agent) or of irradiation on testicular immunology is presented. The few data available on this topic were mainly acquired in rodents. Overall, the manuscript is interesting, clear and well presented. However, modifications of the manuscript are required.

1. The authors should precise in the entire manuscript at what age rodents and humans were treated by chemotherapy or radiotherapy. Were the treatments administered during the prepubertal period or adulthood? 

2. The authors mainly conclude on the effects of hormone suppression in the conclusion section. Please conclude on the impact of cancer treatment on testicular immunology instead. 

3. Several sentences/words are not clear. Please rephrase:

- Lines 17-18: “rodent data on immunological differences in chemotherapy or radiotherapy including few humans involved”

- Line 35: “cell death via apoptosis, e.g., through activated T cells in inflammation”

- Lines 122-123: “infertility after busulfan is in a time- and dose-dependent process and may be related to different species or ages of mice”

- Line 128: “changes to spermatogonia”

- Line 177: “other drugs”

- Line 182: “same médicine”

- Line 187: “are liked to that in the orchitis testes”

4. Please replace the following words:

- Line 13: Please replace “In particularly” by “In particular”

- Line 28: Please replace “speritubular cells” by “peritubular cells”

- Line 33: Please replace “steroidal molécules”

- Line 51: Please replace “does not produce” by “do not produce”

- Lines 71, 72, 74: Please replace “spermatogonia stem cell” by “spermatogonial stem cell”

- Line 110: Please replace “dropping rapidly to controlled concentrations” by “dropped rapidly to control concentrations”

- Line 126: Please replace “extent testicular injury” by “extent of testicular injury”

- Line 160: Please replace “lumens” by “lumen”

5. In Table 1:

- Please replace “tansformiing growth factor” by “transforming growth factor”

- Please replace “TNF a” by “TNFa”

- Is there an increase in ZO-1, occludin and claudin-11 after irradiation and in autoimmune orchitis? (see lines 157-158)

- Please replace “occluding” by “occludin” (also in Figure 1)

- Please remove “protein s”

- Please replace “testosteron” by “testosterone”

- Please replace “contractillity” by “contractility”

- Please replace “T cell-mediated immune” by “T cell-mediated immune response”

Author Response

Responses to Reviewer 1’s comments:

In this review, the impact of the administration of busulfan (a chemotherapeutic agent) or of irradiation on testicular immunology is presented. The few data available on this topic were mainly acquired in rodents. Overall, the manuscript is interesting, clear and well presented. However, modifications of the manuscript are required.

Comments1: The authors should precise in the entire manuscript at what age rodents and humans were treated by chemotherapy or radiotherapy. Were the treatments administered during the prepubertal period or adulthood?

Answer: We agree with the comments and added the age of rodents and humans were treated by chemotherapy or radiotherapy in the entire manuscript(see lines 81; lines 86; lines 90-91; lines 134; lines 137; lines 141-142; lines 144; lines 147; lines 151; lines 152; lines 154; lines 173 and lines 180-181). 

Comments2: The authors mainly conclude on the effects of hormone suppression in the conclusion section. Please conclude on the impact of cancer treatment on testicular immunology instead.

Answer: Thanks for your comments, we have corrected it and conclude on the impact of testicular immunology instead by cancer treatment as “From the above limited reported data, the impaired reproductive functions induced by cancer treatment including chemotherapy and radiotherapy are related with the different immune-pathophysiological conditions. Especially, with the breakdown of BTB by irradiation, germ cell autoantigens inside BTB might leak out repeatedly, leading to continuous supply of autoantigens for immune stimulation with resultant ASA production and prolongation of the testicular inflammation. It is well known that the mechanisms of immune privilege in male reproductive organs are still far from complete understanding. The mechanisms underlying the autoimmune inflammatory response and the mechanisms of orchitis have been broadly investigated and comprehensively reviewed in rodent and human. Although damage to and recovery of spermatogenesis after cancer treatment have been extensively studied, there is little information regarding the role of differences in testicular immunology in cancer treatment-induced male infertility.Presently, the information on therapy for cancer treatment-induced male infertilityis also limited” in Conclusion (see lines 199-211).

Comments3: Several sentences/words are not clear. Please rephrase:

- Lines 17-18: “rodent data on immunological differences in chemotherapy or radiotherapy including few humans involved”

Answer: Thanks for your comments, we have corrected it to “rodent and human data on immunological differences in chemotherapy or radiotherapy” (see lines 17-18).

- Line 35: “cell death via apoptosis, e.g., through activated T cells in inflammation”

Answer: Thanks for your comments, we have corrected it to “induces cell death via apoptosis, such as activated T cells in inflammation” (see lines 35).

- Lines 122-123: “infertility after busulfan is in a time- and dose-dependent process and may be related to different species or ages of mice”

Answer: Thanks for your comments, we have corrected it to “that the infertility after busulfan is in a time- and dose-dependent manner, and is also possibly relate with different species or ages of mice” (see lines 123-125).

- Line 128: “changes to spermatogonia”

Answer: Thanks for your comments, we have corrected it to “changes in spermatogonia” (see lines 130).

- Line 177: “other drugs”

Answer: Thanks for your comments, we have corrected it to “whereas administered an oriental medicine” (see lines 182).

- Line 182: “same médicine”

Answer: Thanks for your comments, we have corrected it to “the above oriental medicine” (see lines 187).

- Line 187: “are liked to that in the orchitis testes”

Answer: Thanks for your comments, we have corrected it to “are similarly to that in the orchitis testes” (see lines 192).

Comments4: Please replace the following words:

- Line 13: Please replace “In particularly” by “In particular”

- Line 28: Please replace “speritubular cells” by “peritubular cells”

- Line 33: Please replace “steroidal molécules”

- Line 51: Please replace “does not produce” by “do not produce”

- Lines 71, 72, 74: Please replace “spermatogonia stem cell” by “spermatogonial stem cell”

- Line 110: Please replace “dropping rapidly to controlled concentrations” by “dropped rapidly to control concentrations”

- Line 126: Please replace “extent testicular injury” by “extent of testicular injury”

- Line 160: Please replace “lumens” by “lumen”

Answer: Thanks for your kind advices and we corrected the above words carefully (see lines 13; lines 28; lines 33; lines 51; lines 71,72,74; lines 111; lines 128 and lines 165).

Comments5:In Table 1:

- Please replace “tansformiing growth factor” by “transforming growth factor”

- Please replace “TNF a” by “TNFa”

Answer: Thanks for your kind advices and we have corrected the above words carefully (see Table 1).

- Is there an increase in ZO-1, occludin and claudin-11 after irradiation and in autoimmune orchitis? (see lines 157-158)

Answer: Thank you for your attention and we have corrected its to decresed arrow in Table 1.

- Please replace “occluding” by “occludin” (also in Figure 1)

- Please remove “protein s”

- Please replace “testosteron” by “testosterone”

- Please replace “contractillity” by “contractility”

- Please replace “T cell-mediated immune” by “T cell-mediated immune response”

Answer: Thanks for your kind advices and we have corrected the above words carefully (see Table 1).

Reviewer 2 Report

The mechanisms of immune privilege in both male and female reproductive organs are still far from complete understanding. The authors of the submitted manuscript have undertaken even more challenging issue, trying to get together the known facts regarding the immune aspects  of impaired spermatogenesis that result from cancer treatment, mainly based on the rodent models.

The authors cite the number of facts, however, research data reported so far are rather limited. The available studies are designed in different ways and in different models, so comparison of the results is not easy. For these reasons the submitted manuscript is rather a compilation of the reported data than their critical review.

I appreciate the work the authors paid for literature review and data acquisition, however, in my opinion the available material is too limited to reach any solid conclusions. The text may be helpful for the design of further research in this particular area, but this could be interesting only for a narrow group of specialists.

I do not find the manuscript informative for a wide spectrum of readers, so I cannot recommend the article for publication in the International Journal of Molecular Sciences.

Author Response

Responses to Reviewer 2’s comments:

The mechanisms of immune privilege in both male and female reproductive organs are still far from complete understanding. The authors of the submitted manuscript have undertaken even more challenging issue, trying to get together the known facts regarding the immune aspects of impaired spermatogenesis that result from cancer treatment, mainly based on the rodent models.

Comments1. The authors cite the number of facts, however, research data reported so far are rather limited. The available studies are designed in different ways and in different models, so comparison of the results is not easy. For these reasons the submitted manuscript is rather a compilation of the reported data than their critical review.

Answer: Thanks for your comments. As we have written in the manuscript, the growing population of young cancer survivors and the trend toward postponing pregnancy until later in life have shifted researchers’ focus toward understanding treatment-induced sequelae, particularly the effects of cancer and/or treatment on fertility. However, there is little information regarding the role of differences in testicular immunology in cancer treatment-induced male infertility. From the limited reported data, we marshaled the impaired reproductive functions induced by cancer treatment including chemotherapy and radiotherapy are related with the different immune-pathophysiological conditions. Especially, with the breakdown of BTB by irradiation, germ cell autoantigens inside BTB might leak out repeatedly, leading to continuous supply of autoantigens for immune stimulation with resultant ASA production and prolongation of the testicular inflammation. The knowledge of this immunopathologic microenvironment will be useful to understand infertility and gonadal dysfunction as adverse effects of anticancer therapy and to lead to an availability of therapy on male infertility after cancer treatment.

Comments2. I appreciate the work the authors paid for literature review and data acquisition, however, in my opinion the available material is too limited to reach any solid conclusions. The text may be helpful for the design of further research in this particular area, but this could be interesting only for a narrow group of specialists.I do not find the manuscript informative for a wide spectrum of readers, so I cannot recommend the article for publication in the International Journal of Molecular Sciences.

AnswerThanks for your comments and we believe that this review will be helpful for the design of further research inreproductive area.As be written in “The Special Issue Infromation” of Reproductive ImmunologySpecial Issue in the International Journal of Molecular Sciences, although highly developed in vitro fertilization techniques are available, many couples still face the problem of childlessness. And to lead to an availability of therapy on male infertility after cancer treatment, elucidation of the immunological mechanisms underlying aspermatogenesis by cancer treatment could be helpful. 

Reviewer 3 Report

The article entitled „Differences in cancer treatment-induced male infertility: the role of testicular immunology” is well written.

Despite the mini review presented herein is very well written, the title seems to general for the content. The article mainly summarizes the differences between busulfan treatment (chemotherapy) and radiotherapy. For this reason, I would suggest changing the title to be more precise toward content.

I have one question regarding the language used as formulation: “including few humans involved.” In the last sentence of Abstract seems odd for me.

The other suggestion is to improve Figure 1. I feel that spermatogonial cells could be better distinguished visually in the figure to improve self-describing. As the action toward spermatogonial cells are one of the most important differences between chemo and radio therapy it could be the point of great importance for the audience.

I didn’t find any mistakes in the references.

Overall I suggest to accept the article as mini-review but with a title more specific to the content.

The article entitled „Differences in cancer treatment-induced male infertility: the role of testicular immunology” is well written.

Despite the mini review presented herein is very well written, the title seems to general for the content. The article mainly summarizes the differences between busulfan treatment (chemotherapy) and radiotherapy. For this reason, I would suggest changing the title to be more precise toward content.

I have one question regarding the language used as formulation: “including few humans involved.” In the last sentence of Abstract seems odd for me.

The other suggestion is to improve Figure 1. I feel that spermatogonial cells could be better distinguished visually in the figure to improve self-describing. As the action toward spermatogonial cells are one of the most important differences between chemo and radio therapy it could be the point of great importance for the audience.

I didn’t find any mistakes in the references.

Overall I suggest to accept the article as mini-review but with a title more specific to the content.

Author Response

Responses to Reviewer 3’s comments:

The article entitled “Differences in cancer treatment-induced male infertility: the role of testicular immunology” is well written.

Comments1. Despite the mini review presented herein is very well written, the title seems to general for the content. The article mainly summarizes the differences between busulfan treatment (chemotherapy) and radiotherapy. For this reason, I would suggest changing the title to be more precise toward content.

Answer: Thanks for your kind advices, we have changed the title to be “Effect of chemotherapy and radiotherapy on spermatogenesis: the role of testicular immunology”.

Comments2. I have one question regarding the language used as formulation: “including few humans involved.” In the last sentence of Abstract seems odd for me.

Answer: Thanks for your comments, we have corrected it to “rodent and human data on immunological differences in chemotherapy or radiotherapy” (see lines 17-18).

Comments3. The other suggestion is to improve Figure 1. I feel that spermatogonial cells could be better distinguished visually in the figure to improve self-describing. As the action toward spermatogonial cells are one of the most important differences between chemo and radio therapy it could be the point of great importance for the audience.

Answer: Thanks for your kind advices and we improved Figure 1 to distinguish the most important differences, such as spermatogonial cells, BTB and testicular macrophage, more visually between chemo and radio therapy.. 

Round  2

Reviewer 2 Report

The improvement of the "Conclusion" part is appreciable, as well as more detailed description of animals used in the studies.

The manuscript still needs text editing:

l.78: with regard or regarding

l.81: germ cell apoptosis

l. 92: 8-18 weeks age - what? - mice? probably 8-10 weeks old mice would be better

l.125: is also possibly related .

Author Response

Responses to Reviewer 2’s comments:

 The improvement of the "Conclusion" part is appreciable, as well as more detailed description of animals used in the studies.

Comments: The manuscript still needs text editing:

- L.78: with regard or regarding

- L.81: germ cell apoptosis

- L.92: 8-18 weeks age - what? - mice? probably 8-10 weeks old mice would be better

- L.125: is also possibly related

Answer: Thanks for your kind advices and we corrected the above words carefully (see lines 78; lines 81; lines 92 and lines 125).

 Thank you again for your comments.